# COVID-19 Pandemic: Knowledge and Attitudes in Public Markets in the Former Katanga Province of the Democratic Republic of Congo

**DOI:** 10.3390/ijerph17207441

**Published:** 2020-10-13

**Authors:** Trésor Carsi Kuhangana, Caleb Kamanda Mbayo, Joseph Pyana Kitenge, Arlène Kazadi Ngoy, Taty Muta Musambo, Paul Musa Obadia, Patrick D.M.C. Katoto, Célestin Banza Lubaba Nkulu, Benoit Nemery

**Affiliations:** 1Department of Public Health, Faculty of Medicine and Public Health, University of Kolwezi, Kolwezi, Democratic Republic of the Congo; 2Unit of Toxicology and Environment, School of Public Health, Faculty of Medicine, University of Lubumbashi, Lubumbashi, Democratic Republic of the Congo; kamandacaleb@gmail.com (C.K.M.); arlene.kazadingoy@gmail.com (A.K.N.); tatymuta17@gmail.com (T.M.M.); musa.p.obadia@gmail.com (P.M.O.); clubabankulu2017@gmail.com (C.B.L.N.); 3Higher School of Industrial Engineers, University of Lubumbashi, Lubumbashi, Democratic Republic of the Congo; 4Occupational Medicine and Environmental Health, Department of Public Health, Faculty of Medicine, University of Lubumbashi, Lubumbashi, Democratic Republic of the Congo; docteurpyana@gmail.com; 5Haut-Katanga Provincial Inspection of Health, Ministry of Public Health, Lubumbashi, Democratic Republic of the Congo; 6Haut-Katanga Provincial Division of Health, Ministry of Public Health, Lubumbashi, Democratic Republic of the Congo; 7Department of Internal Medicine, Division of Respiratory Medicine & Prof. Lurhuma Biomedical Research Laboratory, Mycobacterium Unit, Catholic University of Bukavu, Bukavu, Democratic Republic of the Congo; katotopatrick@gmail.com; 8Department of Medicine and Center for Infectious Diseases, Faculty of Medicine and Health Sciences, Stellenbosch University, Cape Town 7505, South Africa; 9Centre for Environment and Health, Department of Public Health and Primary Care, KU Leuven, 3000 Leuven, Belgium

**Keywords:** SARS-CoV-2, prevention and control, mass education, population health, Democratic Republic of Congo, Africa

## Abstract

*Background.* Public markets were exempted from the restrictive regulations instituted to limit the rapid spread of the COVID-19 pandemic in the Democratic Republic of the Congo (DRC). In the early stage of the pandemic, we assessed people’s knowledge, attitudes, and behavior on public markets towards COVID-19. *Methods*. We conducted a cross-sectional study from 16 to 29 April 2020 among sellers and customers frequenting the food sections of ten public markets in three large cities (Kolwezi, Likasi, and Lubumbashi) and one small town (Lwambo) of the former Katanga province. We administered a questionnaire on knowledge (about clinical characteristics, transmission and prevention) and on attitudes in relation to COVID-19. We also observed prevailing practices (hand-washing and mask-wearing). *Results*: Of the 347 included participants (83% women, 83% sellers), most had low socioeconomic status and a low level of education. Only 30% of participants had correct knowledge of COVID-19. The majority of the respondents (88%) showed no confidence in the government’s ability to manage the upcoming pandemic crisis. Nearly all respondents (98%) were concerned about the associated increase in food insecurity. Preventive practices were rarely in place. *Conclusion*: For an effective implementation of measures to prevent the spread of COVID-19 in Africa, appropriate health education programs to improve knowledge and attitudes are warranted among the population frequenting public markets.

## 1. Introduction

In December 2019, coronavirus disease 2019 (COVID-19), caused by severe acute respiratory syndrome coronavirus 2 (SARS-CoV-2), started in Wuhan, in the Hubei province of China [1]. From the onset of the pandemic, it was feared that Sub-Saharan Africa would not be spared because of the high volume of air traffic and trade between China and Africa [2]. The high prevalence of infectious diseases, coupled with weak healthcare systems in Africa, might lead to high mortality rates because of co-morbidity in populations [2,3], but it could also be assumed, on the other hand, that a younger African population distribution would lessen the death rate of COVID-19 on the continent, since mortality rates were generally high in older people [4].

When COVID-19 was declared a pandemic by WHO on March 11, 2020 [5], a total of 47 confirmed COVID-19 cases, including 28 (60%) imported cases, with no deaths, had been reported across nine countries, including the Democratic Republic of Congo (DRC), with one case in its capital Kinshasa [6].

Even if some Sub-Saharan African countries installed a total lockdown for a long time (e.g., South Africa), many Sub-Saharan African countries did not or were not able to implement lasting draconian lockdown measures because these would too severely affect social and economic systems. In search of income for the day-to-day livelihood of extended families, many Africans were forced to fend for survival and ignore concerns about contracting COVID-19 [3,7,8].

Around one week after COVID-19 was declared a pandemic, the government of the DRC closed schools, churches, bars, and nightclubs and banned all gatherings involving more than 20 people in public places. People were asked to wash hands with hydro-alcoholic solution or water and soap, to avoid close contacts and maintain social distancing, wear a face mask, no longer greet each other by shaking hands, and to call one of the “COVID-19 attack teams” telephone numbers in case of a suspected case [9]. However, public markets and shops selling food were exempted from these lockdown regulations. As a result, the markets remained crowded and, hence, potential high-risk places for the transmission of SARS-CoV-2.

The population’s behavior has been associated with the evolution of the COVID-19 pandemic [10]. In the current study done during the early stage of the COVID-19 pandemic, we aimed to assess awareness of the risk of contamination among people frequenting public markets in the former Katanga province in the south of the DRC.

## 2. Materials and Methods

Between 16 and 29 April 2020, we conducted a cross-sectional study among adult people (>18 years) frequenting ten public markets (mainly selling food) in the former Katanga province. Nine markets were located in three large industrial cities: two markets in Lubumbashi (about 2 million inhabitants) and three markets in Likasi (about 450,000 inhabitants) in the Haut-Katanga province, and four markets in Kolwezi (about 450,000 inhabitants) in the Lualaba province; one market was located in Lwambo (25,000 inhabitants), a small mining town north of Likasi. Authorizations from the local administrative authorities were obtained before the surveys. All participants agreed to participate in the study and the study was approved by the Committee of Medical Ethics of the University of Lubumbashi (UNILU) (UNILU/CENI/223/2020), provided that individual confidentiality would be guaranteed.

The questionnaire was elaborated in French, translated into Swahili and back-translated to check for accuracy. The questions were based on the then-available scientific information about COVID-19 and concerned simple sociodemographic characteristics and 20 items to assess knowledge about clinical characteristics, transmission and prevention, and attitudes in relation to COVID-19. The questionnaire was briefly piloted among the research team and their relatives. For the actual surveys, paper questionnaires were administered face-to-face by teams of three trained interviewers who were present during around 5 h on each of the selected markets. They approached potential adult participants, either market vendors or customers, explaining that they were academic researchers (i.e., not governmental employees) doing a survey about COVID-19; anonymity of the findings was guaranteed and participants gave oral consent to respond. No attempts were made to obtain samples having identical sociodemographic characteristics in the various locations.

To obtain information on behavior and good practices against the dissemination of SARS-CoV-2 (such as hand-washing and mask-wearing), the interviewers also checked (unobtrusively) a number of observed practices on the individual questionnaire forms and they further noted general observations in logbooks. Photographs were also made, with care being taken not to make recognizable full-face photographs.

To estimate socioeconomic status (SES), we used, as a proxy, the main reported source of information regarding COVID-19: participants who declared having obtained information from watching their own televisions or through the internet (smartphone) were classified as having a “moderate” income; participants who obtained information from their radio were considered as having a low income; participants who did not mention the above sources of information and/or received information only from “other people” were considered as having a very low income.

When a participant responded “I don’t know”, this was assimilated to a “no” or “wrong” answer. We used SPSS V.20 for statistical analysis, using Chi-square test and Cramer’s V test as appropriate. The threshold for statistical significance was set at *p* < 0.05 (two-sided).

## 3. Results

Our results are based on data obtained from a total of 347 persons present, as sellers or buyers, on 10 markets surveyed between 16 and 29 April 2020. We did not always obtain complete data from each respondent, due to occasional refusals or inability to respond to some questions, as well as a number of incompletely or poorly filled out questionnaires in view of the field conditions. The survey was stopped just after the first positive case of COVID-19 had been reported in the former Katanga province.

Table 1 summarizes the sociodemographic characteristics of the study participants. The majority were food vendors (83%) and women (83%). The mean age was 37.4 years (SD 10.7; range 18–70 years). Of note, age was available for only 233 respondents (67%) because it is unusual, for cultural reasons, to disclose one’s age in this region. In general, participants differed across the four main locations with regards to gender distribution, age, socioeconomic position, and the proportion of sellers versus buyers. Based on their main source of information, half the participants could be considered as having a moderate income (access to TV or internet) and the other half as having a low income (access to radio) or a very low income. Sellers and buyers did not differ by age [37.2 (SD 10.8) years and 38.2 (SD 10.6) years, respectively], but they differed somewhat with regard to gender (15% of sellers being males versus 29% among buyers, *p* < 0.05) and SES (66% of buyers having a moderate income versus 48% among sellers, *p* < 0.05).

Considering answers to the knowledge questions (Table 2), two-thirds of the respondents did not know about COVID-19 and 60% answered that the COVID-19 pandemic was not present in the DRC, despite the 400 cases reported at the time of the surveys in Kinshasa, the capital city, admittedly located at more than 1500 km from the study area. The majority of participants (65%) were unaware of the possible COVID-19 transmission by asymptomatic patients. On the other hand, about 60% of the respondents considered COVID-19 as having the same severity as malaria or Ebola disease (an outbreak of which was ongoing in the country, but also at a distance of more than 1500 km). Although statistical comparisons between the various locations revealed significant differences for many questions, no systematic location-specific pattern emerged with regard to knowledge.

Interestingly, in Lubumbashi the market surveys were conducted after 23 April, i.e., after a first case of COVID-19 had been allegedly registered in the city [11], yet 72% of respondents did not believe that the disease was already in Lubumbashi.

With regards to attitude, our results indicate that 13% of the respondents were prepared to visit a COVID-19-infected neighbor or colleague, and 31% would visit an infected family member (child, spouse, etc.). A substantial proportion (40%) were convinced that COVID-19 had a “demonic” origin, and 16% believed the virus had been bioengineered in laboratories. Regarding food security, the COVID-19 pandemic was felt to have a negative impact by nearly all participants, with 98% reporting a reduction in food quantity and the number of daily meals. Most participants (88%) did not agree that COVID-19 would be successfully controlled and defeated in the DRC. Nearly all respondents (97%) reported using public transportation, which in the area consists of generally cramped small minivans. Again, although statistical comparisons between locations showed significant differences for several questions on attitude, no consistent location-specific pattern could be discerned regarding attitudes.

However, we found associations between responses and individual participants’ characteristics (Table 3). Knowledge and attitudes were weakly to moderately associated with reported sources of information, since correct answers tended to be given more frequently by those stating to have been informed about COVID-19 through medical staff, followed by audiovisual media. Men and women did not appear to differ much in their responses, except that 39% of men (as opposed to 11.5% of women) declared that no preventive actions were required for children and young adults. Overall, sellers and buyers did not differ significantly for most questions on knowledge, although occasional and inconsistent differences did appear between the two groups depending on location (data not shown, available on request). For instance, in Lwambo, sellers tended to have poorer knowledge than buyers, whereas the opposite was the case in Likasi. No significant differences were found between sellers and buyers with regard to attitude.

During the survey, we also systematically recorded general and individual practices against COVID-19. Only five markets had washing devices (an example is shown in Figure 1): in Kolwezi, one market had four communal devices; in Likasi, there was one market with three communal devices and one market with four sellers having their own washing device; in Lubumbashi, one market had one communal device; and no hand-washing facilities were available in Lwambo. However, nowhere did we observe either sellers or buyers washing their hands after reciprocal contacts. Nevertheless, we found weak-to-strong significant associations between almost all questions on knowledge and attitudes and the number of hand-washing devices on the market: the proportions of respondents giving correct answers to knowledge questions and reporting good attitudes were significantly higher in markets with more than two hand-washing facilities than in markets without such facilities (Table 3). Except for only one seller wearing a face mask (in a Lubumbashi market), face masks were not worn, as shown in Figure 2. All payments were by banknotes, and many buyers were observed touching multiple items before choosing one. We also observed that many sellers would blow into plastic bags before placing food into the bags, but we also saw one buyer requesting another plastic bag without prior blowing. In general, social distancing was not respected, although observance of some social distancing was noted for 105 (30%) of the interviewees, with one Kolwezi market standing out in this respect [42/48 (88%) interviewees].

## 4. Discussion

To our knowledge, this is the first survey that attempted to assess public perception in public markets regarding COVID-19 at an early stage of the pandemic in Africa. We administered a simple questionnaire on knowledge and attitudes to people frequenting or working in ten public markets in four different locations in the mining area of the former Katanga province in the southeast of the DRC. Our main findings are that a large proportion of common people had only a poor knowledge of the disease and its prevention, but also that most respondents had little confidence in the ability of the government to tackle the outbreak and that nearly everybody was worried about its consequences on food security rather than on health. Interestingly, positive attitudes seemed to be related to the existence of visible preventive measures, mainly hand-washing facilities, in a market.

We acknowledge upfront the methodological limitations of a survey based on a self-elaborated questionnaire that was administered to a convenience sample without consideration of the size and composition of the target population. In the absence of prior evidence (except from some online studies), formal sample size calculation was not possible, but the numbers of participants were based on practical and time constrains. It should be noted that conditions for performing sophisticated sociological surveys are hardly met in the DRC, and that we wanted to work rapidly, i.e., before the eruption of any major outbreak of the COVID-19 in the area. The questionnaire was based on the then existing scientific literature; it was briefly piloted among members and relatives of the research group. We do not know how many people were present on the chosen markets at the time of the surveys, but they numbered many hundreds. We also did not formally register the total number of people approached to respond to the questionnaire, but we estimate that about one in two invited persons accepted to participate. We also acknowledge that, although the interviewers tried to administer the questionnaire in a standardized way, they often needed to clarify or rephrase the questions. On the other hand, we have no reasons to think that our findings were seriously biased by these methodological shortcomings and we consider that our survey of more than 300 people in ten markets in four different locations may provide a faithful picture of the then-prevailing beliefs and attitudes about COVID-19, not only in the studied area but probably also elsewhere in Africa. Indeed, although variations between respondents in the different locations appeared large for some questions, we were unable to find consistent location-specific patterns regarding knowledge, attitude, or practices and we have, therefore, limited our interpretations to the general picture, as obtained from an analysis of the whole sample.

As expected, the majority (83%) of our participants were women, thus reflecting the gender distribution of jobs and domestic tasks related to food distribution and preparation in Africa. Based on our crude but probably reliable proxy for relative income, our participants were of very-low-to-moderate SES, keeping in mind that even our notion of a “moderate” income must be viewed in the context of a national economy with an average per capita income between 1 and 2 US dollars per day [12]. Although sellers and buyers did vary somewhat by gender and SES, their responses and behavior appeared largely similar, thus supporting again the assumption of representativeness of our cross-sectional sample for the population attending mainly urban markets in this African region.

At the time of our survey, i.e., before COVID-19 had arrived in the area, many respondents proved to have very poor or no knowledge of the characteristics of COVID-19, and this did not correlate significantly with our pragmatic measure of SES. For instance, many respondents did not know the differences between COVID-19 and an endemic condition like malaria or the severe Ebola disease, of which the tenth outbreak was ongoing in the country [13]. Clearly, the majority of the participants were mainly concerned by the increasing food insecurity, i.e., obtaining their “daily food”. This concern was justified since prohibition of moving around would mean loss of income and limited access to food and essential supplies [3,8]. It has been stated that Sub-Saharan African countries cannot afford to maintain stringent lockdown measures over a long period of time [7]. The main information sources reported by the participants were audiovisual media (television, radio) and “other people” in the street, whereas health workers accounted for a very small fraction (4%) of reported information sources. Awareness campaigns, communicated via websites, television, and various social media in the DRC, are usually in French, the education and working language of the DRC, although 22.7% of its people are illiterate (11% among men, 34% among women) [14], and hardly understand French. We insist that health educative programs should provide messages against COVID-19 mainly in local languages.

Our survey indicates that the majority of participants (88%) did not agree that the COVID-19 situation would be under control and defeated in the DRC. A possible explanation for this type of attitude could be the reports of rapid spreading of the disease and the high mortality figures in developed countries, like in Italy and in the United States of America [15]. However, another explanation given by respondents was that “it is hard to trust the Congolese health system since most political authorities usually travel overseas to seek for a better health care”. These findings are in line with data from Bangladesh [16]. They are, however, in contrast with data from Saudi Arabia showing a positive and optimistic attitude toward COVID-19, with 97% of participants convinced that the Saudi government would control the pandemic [17], and with a study done in Malaysia where 96% of participants had confidence that the country would be able to win the battle against COVID-19 [18].

The (limited) presence of hand-washing facilities at the markets seemed to increase the general level of knowledge and to favor adequate attitudes among the surveyed market goers. Nevertheless, even when washing facilities were present, no sellers or buyers were seen washing their hands during our presence in the markets for about five hours per day. This apparent contradiction can be due to a lack of awareness of the need or effectiveness of hand-washing, but it could also be attributed, in part, to the widespread belief that COVID-19 is caused by “demonic forces” (44% in our sample), with the often-expressed implication that “children of God can’t be infected by this disease”. In China, a study found that residents with a relatively high SES, had good knowledge, optimistic attitudes, and appropriate practices towards COVID-19 during the rapid rise period of the outbreak, and good knowledge was associated with optimistic attitudes and appropriate practices towards COVID-19 [19,20]. An online study in the USA reported that, in response to COVID-19, 95.7% of respondents had made changes to their lifestyle, including hand-washing (93.1%), avoidance of social gatherings (89.0%), and stockpiling food and supplies (74.7%) [21].

## 5. Conclusions

It was beyond the scope of this study to address the numerous challenges posed by the COVID-19 pandemic in the DRC [22], but we believed that the objective findings of our pragmatic survey could contribute to offer a holistic approach of its management in Africa [3]. Our study was conducted at a very early stage of the pandemic, i.e., when the first cases were being reported from Kinshasa and before any cases had occurred in the area where the surveys were done (except for the last market surveyed). At that time, it was feared that Sub-Saharan Africa would be severely hit by the pandemic because of the poverty of its population and the resulting difficulties of adopting quarantines, physical distancing, and other barrier methods, on the one hand, and the limited medical and other resources available to manage severe disease, on the other hand. At that early stage of the COVID-19 pandemic in the DRC, we found a generally low awareness among people present, as sellers or buyers, on crowded public markets, and this led us to conclude that targeted health education programs were needed to limit the propagation of COVID-19 in this population with a low level of education. However, it was also felt that further surveys would be needed to assess the behavior dynamic in the general population as the COVID-19 pandemic would evolve in the country. The current findings provide a unique baseline for such studies.

We stand by these early conclusions of our survey that health education programs must be based on a thorough assessment of the population’s knowledge, beliefs, and attitudes. The information given to the public should be provided in understandable and respectful language, and by reliable sources, including health professionals. The international guidelines for infection prevention and control must be adapted at country level, and they should be tailored to the prevailing socioeconomic and public health situations. These considerations are in line with strategies for promoting the adoption of protective behaviors, which have been formulated in an excellent “rapid expert consultation” released by the “Societal Experts Action Network” of the (US) National Academies of Science, Engineering, and Medicine [23].

Since the submission of our article, we have come to realize that the anticipated COVID-19 major catastrophe has not taken place in Africa. Indeed, according to the African Centres for Disease Control and Prevention, by the end of September 2020, the cases of COVID-19 reported for the 55 African Union Member State Countries represented less than 4% of all cases reported globally [24]. In the 47 (mainly Sub-Saharan) countries of the WHO African Region, 1,175,271 cases (and 25,825 deaths) of COVID-19 had been reported by 29 September 2020, with a steady decline since a peak that occurred in July [25]. Moreover, the majority of cases and deaths occurred in a single country, namely South Africa (672, 572 cases, 16,667 deaths) [25]. In the DRC (with a population of 84 million), 10,631 cases and only 272 deaths were reported [25]. The capital Kinshasa was the epicenter, with 77% of the total cases [26,27], and only 432 confirmed cases were reported from the two provinces where our survey took place [27]. Although COVID-19 cases undoubtedly escaped recognition, this is not the sole explanation for these relatively low numbers, and the reasons for the hitherto low incidence of COVID-19 in Sub-Saharan Africa compared to other regions in the world have not been fully elucidated. Public health interventions, such as travel bans and early lockdowns, may have played a role, but sociodemographic characteristics, such as a younger population, low population density and mobility, as well as cross-immunity with other infections may have contributed [24]. The propagation of SARS-CoV-2 within the general population may also have been less efficient in Sub-Saharan Africa than in more affluent, industrially developed, and highly urbanized areas, because African people spend more time outdoors, not so much because of a “better climate”, but because of the prevailing socioeconomic circumstances. In other words, it is conceivable that the very need, for most people, to purchase food from open markets (rather than from closed supermarkets) may actually have mitigated the spread of the virus. In addition, and perhaps of equal or more importance than time spent outdoors, it is plausible that building characteristics play a substantial role in viral transmission. In industrially developed countries, people not only spend a large proportion of their time indoors, they generally do so in air-tight buildings with much less natural ventilation than in Africa. This applies to home dwellings and communal buildings, such as workplaces, classrooms, shops, hospitals, churches, restaurants, and bars, and even to public transport vehicles. In this way, an affluent society living in “high-quality” buildings may, paradoxically, have been more susceptible to the pandemic than a poor society with relatively open, well-ventilated constructions (sometimes without windowpanes). These hypotheses need to be verified, but if they are confirmed, this should obviously not lead to considering poverty as an acceptable remedy against COVID-19.

## Figures and Tables

**Figure 1 ijerph-17-07441-f001:**
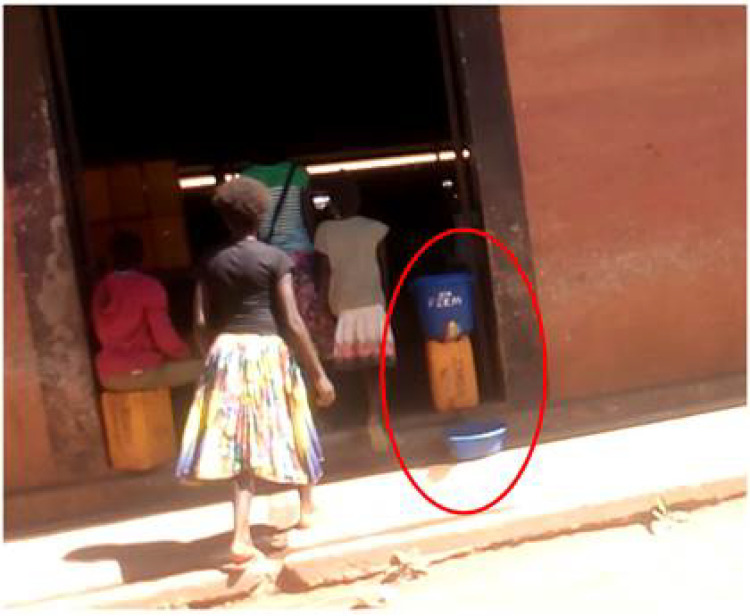
Hand-washing device (in red solid circle) in a public market in Lubumbashi, Democratic Republic of Congo.

**Figure 2 ijerph-17-07441-f002:**
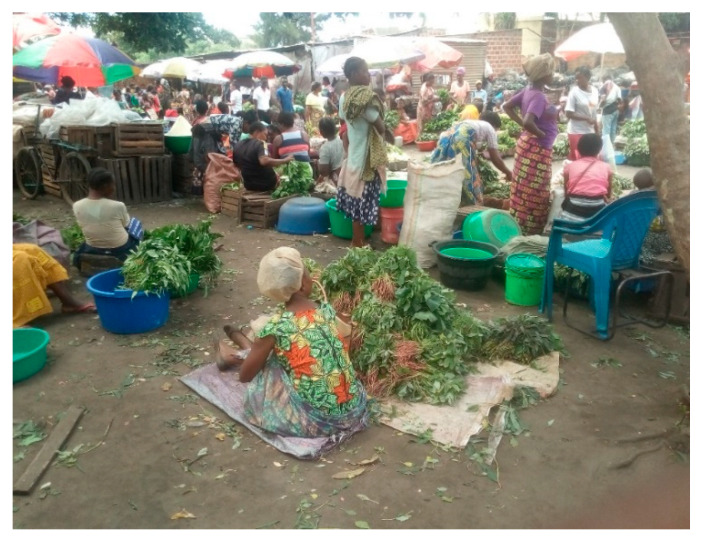
Food section in a public market in Kolwezi, Democratic Republic of Congo.

**Table 1 ijerph-17-07441-t001:** Sociodemographic characteristics of the people included in questionnaire surveys done between 16 and 29 April 2020 on ten markets in four locations of the former Katanga province of DR Congo.

	Kolwezi	Lwambo	Likasi	Lubumbashi	*p*	Total
Dates of surveys	16–17 April	18 April	19–20 April	24–29 April		
Number of markets, *n*	4	1	3	2		**10**
Number of respondents, *n* (% of total)	114 (32.9)	64 (18.4)	80 (23.1)	89 (25.6)		**347 (100)**
Female, *n* (%)	95 (83.3)	46 (71.9)	75 (93.8)	72 (80.9)	<0.01	**288 (83.0)**
Male, *n* (%)	19 (16.7)	18 (28.1)	5 (6.2)	17 (19.1)		**59 (17.0)**
Age, years, mean (SD)	40.7 (11.6)	38.1 (12.4)	36.2 (9.6)	36.8 (10.6)	0.168	**37.4 (10.7)**
Sellers, *n* (%)	102 (89.5)	41 (64.1)	67 (83.8)	78 (87.6)	<0.0001	**288 (83.0)**
Buyers, *n* (%)	12 (10.5)	23 (35.9)	13 (16.2)	11 (12.4)		**59 (17.0)**
Socioeconomic status, *n* (%)						
Moderate income	56 (49.1)	48 (75.0)	38 (47.5)	35 (39.3)		**177 (51.0)**
Low income	25 (21.9)	8 (12.5)	30 (37.5)	23 (25.9)	<0.0001	**86 (24.8)**
Very low income	33 (29.0)	8 (12.5)	12 (15.0)	31 (34.8)		**84 (24.2)**
Information sources, *n* (%)						
Audiovisual media	83 (72.8)	51 (79.7)	57 (21.7)	72 (80.9)		**263 (75.8)**
Medical staff	7 (6.1)	0 (00.0)	6 (46.2)	0 (00.0)	0.077	**13 (3.7)**
People	24 (21.1)	13 (20.3)	17 (23.9)	17 (19.1)		**71 (20.5)**

Data are numbers with percentages. Data in bold (last column) concern the total participants. Socioeconomic status (SES) was based on the reported main source of information received on COVID-19: participants who declared having obtained information from watching their own televisions or through the internet (smartphone) were classified as having a moderate income; participants who obtained information from their radio were considered as having a low income; participants who did not mention these sources of information and/or received information only from “other people” were considered as having a very low income. DR: Democratic Republic.

**Table 2 ijerph-17-07441-t002:** Responses of participants to questionnaire surveys done between 16 and 29 April 2020 on ten markets in four locations of the former Katanga province of DR Congo.

	Kolwezi	Lwambo	Likasi	Lubumbashi	*p* (χ^2^)	Total
**Knowledge (K)**						
*K1. COVID-19 is a viral disease caused by the SARS-COV-2, the main clinical symptoms being fever, fatigue, dry cough, and myalgia.*						
Yes	16 (14.0)	21 (32.8)	51 (63.7)	27 (30.3)	<0.001	**115 (33.1)**
No	98 (86.0)	43 (67.2)	29 (36.2)	62 (69.7)		**232 (66.9)**
*K2. Do you think this disease exists in the DRC?*						
Yes	53 (46.5)	23 (35.9)	32 (40.0)	30 (33.7)	0.271	**138 (39.8)**
No	61 (53.5)	41 (64.1)	48 (60.0)	59 (66.3)		**209 (60.2)**
*K3. Do you know the modes of transmission of COVID-19?*						
Yes	69 (60.5)	24 (37.5)	46 (57.5)	40 (44.9)	<0.05	**179 (51.6)**
No	45 (39.5)	40 (62.5)	34 (42.5)	49 (55.1)		**168 (48.4)**
*K4. Can someone with COVID-2019 who does not have a fever contaminate his or her contacts?*						
Yes	72 (63.2)	15 (23.4)	32 (40.0)	34 (38.2)	0.161	**123 (35.4)**
No	42 (36.8)	49 (76.6)	48 (60.0)	55 (61.8)		**224 (64.6)**
*K5. Do you know the preventive measures against COVID-1?*						
Yes	88 (77.2)	21 (32.8)	36 (45.0)	72 (80.9)	<0.001	**217 (62.5)**
No	26 (22.8)	43 (67.2)	44 (55.0)	17 (19.1)		**130 (37.5)**
*K6. In your opinion, COVID-19 comes from:*						
China	86 (75.4)	24 (37.5)	46 (57.5)	67 (75.3)	<0.001	**223 (64.3)**
Others	28 (24.6)	40 (62.5)	34 (42.5)	22 (24.7)		**124 (35.7)**
*K7. In your opinion, COVID-19 affects:*						
Everyone	67 (58.8)	44 (68.8)	76 (95.0)	75 (84.3)	<0.001	**262 (75.5)**
Especially white people	44 (38.6)	15 (23.4)	4 (5.0)	1 (1.1)		**64 (18.4)**
Others	3 (2.6)	5 (7.8)	0 (0.0)	13 (14.6)		**21 (6.1)**
*K8. According to you, compared to malaria, COVID-19 is:*						
Less severe	25 (21.9)	3 (4.7)	0 (0.0)	4 (4.5)	<0.001	**32 (9.2)**
As severe	29 (25.4)	37 (57.8)	80 (100)	60 (67.4)		**206 (59.4)**
More severe	60 (52.6)	24 (37.5)	0 (0.0)	25 (28.1)		**109 (31.4)**
*K9. In your opinion, compared to EBOLA, COVID-19 is:*						
Less severe	29 (25.4)	10 (15.6)	0 (0.0)	2 (2.2)		**41 (11.8)**
As severe	33 (28.9)	40 (62.5)	80 (100)	60 (67.4)	<0.001	**213 (61.4)**
More severe	41 (36.0)	14 (21.9)	0 (0.0)	24 (27.0)		**79 (22.8)**
I don’t know	11 (9.6)	0 (0.0)	0 (0.0)	3 (3.4)		**14 (4.0)**
*K10. In your opinion, is COVID-19 already in Lubumbashi? **						
Yes				18 (28.1)		**18 (28.1)**
No				46 (71.9)		**46 (71.9)**
**Attitude (A)**						
*A1. Can you visit someone who has COVID-19?*						
Yes	21 (18.4)	8 (12.5)	12 (15.0)	3 (3.4)	<0.05	**44 (12.7)**
No	93 (81.6)	56 (87.5)	68 (85.0)	86 (96.6)		**303 (87.3)**
*A2. If it is someone in your family (child, spouse...) who suffers from COVID-19, can you visit him/her?*						
Yes	48 (42.5)	29 (45.3)	21 (26.2)	9 (10.1)	<0.001	**107 (30.9)**
No	65 (57.5)	35 (54.7)	59 (73.8)	80 (89.9)		**239 (69.1)**
*A3. The people most affected by COVID-19 are adults. Do you think there is a need to take action to prevent COVID-19 infection for children and young adults?*						
Yes, it is necessary for them	86 (75.4)	54 (84.4)	80 (100.0)	71 (79.8)	<0.001	**291 (83.9)**
No, it is not necessary for them	28 (24.6)	10 (15.6)	0 (0.0)	18 (20.2)		**56 (16.1)**
*A4. Do you agree that COVID-19 will be successfully controlled and defeated in the DRC?*						
Yes	9 (7.9)	6 (9.4)	15 (18.8)	12 (13.5)	0.120	**42 (12.1)**
No	105 (92.1)	58 (90.6)	65 (81.2)	77 (86.5)		**305 (87.9)**
*A5. How do you find the origin of COVID-19?*						
Normal	10 (8.8)	27 (42.2)	6 (7.5)	15 (16.9)	<0.001	**58 (16.7)**
Demonic	66 (57.9)	8 (12.5)	28 (35.0)	39 (43.8)		**141 (40.6)**
Made	22 (19.3)	15 (23.4)	7 (8.8)	14 (15.7)		**58 (16.7)**
I don’t know	16 (14.0)	14 (21.9)	39 (48.8)	21 (23.6)		**90 (25.9)**
*A6. During this COVID-19 crisis period, do you eat as before?*						
Yes	7 (6.1)	0 (0.0)	0 (0.0)	0 (0.0)	<0.005	**7 (2.0)**
No	107 (93.9)	64 (100.0)	80 (100.0)	89 (100.0)		**340 (98.0)**
*A7. What type of transport do you use to get to the market?*						
Private transportation	6 (5.3)	5 (7.8)	0 (0.0)	0 (0.0)	<0.01	**11 (3.2)**
Public transportation	108 (94.7)	59 (92.2)	80 (100.0)	89 (100.0)		**336 (96.8)**
*A8. How often do you go to the market for supplies or for selling per week?*						
1–4 days	1 (0.9)	0 (0.0)	0 (0.0)	3 (3.4)	0.133	**4 (1.2)**
5–7 days	113 (99.1)	64 (100)	80 (100)	86 (96.6)		**343 (98.8)**
*A9. If someone around you has symptoms of COVID-19, what would you do?*						
I don’t know	0 (0.0)	4 (6.2)	11 (13.8)	13 (14.6)	<0.001	**28 (8.1)**
We will take him to the hospital	33 (28.9)	5 (7.8)	4 (5.0)	12 (13.5)		**54 (15.6)**
We will call the COVID-19 attack team	81 (71.1)	55 (85.9)	65 (81.2)	64 (71.9)		**265 (76.4)**
*A10. If you think you have COVID-19, what would you do?*						
I don’t know	0 (0.0)	4 (6.2)	10 (12.5)	13 (14.6)	<0.001	**27 (7.8)**
I’ll go to the hospital	41 (36.0)	4 (6.2)	10 (12.5)	11 (12.4)		**66 (19.0)**
I will call the COVID-19 attack team	73 (64.0)	56 (87.5)	60 (75.0)	65 (73.0)		**254 (73.2)**

Data are shown as *n* (percentage), with the answer “I don’t know” assimilated to the answer “No”. Data in bold (last column) concern the total participants. * K10 Question added on 04/25/2020 after a first case of COVID-19 was registered in Lubumbashi.

**Table 3 ijerph-17-07441-t003:** Association between personal characteristics and responses to questionnaire surveys and observations made between 16 and 29 April 2020 on ten markets in four locations of the former Katanga province of DR Congo.

	Number of Hand-Washing Devices on Market (0 to 4)	Socioeconomic Status (Moderate, Low, Very Low)	Information Sources (Medical Staff, Audiovisual Media, Other People)	Gender (Male, Female)
	*X*^2^, df, *p* (Cramer’s V)	*X*^2^, df, *p* (Cramer’s V)	*X*^2^, df, *p* (Cramer’s V)	*X*^2^, df, *p* (Cramer’s V)
*K1. COVID-19 is a viral disease caused by the COVID-19, the main clinical symptoms being fever, fatigue, dry cough, and myalgia.*	57.6, 4, <0.01	0.7, 2, 0.69	11.1, 2, <0.01	0.2, 1, 0.66
	(0.41) ***	(0.05)	(0.18) *	(0.02)
*K2. Do you think this disease exists in the DRC?*	17.0, 4, <0.01	3.4, 2, 0.18	10.3, 2, <0.01	1.8, 1, 0.19
	(0.22) **	(0.10)	(0.17) *	(0.07)
*K3. Do you know the modes of transmission of COVID-19?*	61.1, 4, <0.01	0.3, 2, 0.88	20.3, 2, <0.01	1.7, 1, 0.19
	(0.42) ***	(0.03)	(0.24) **	(0.07)
*K4. Can someone with COVID-2019 who does not have a fever contaminate his or her contacts?*	68.9, 4, <0.01	2.7, 2, 0.25	11.4, 2, <0.01	0.3, 1, 0.57
	(0.45) ***	(0.90)	(0.18) *	(0.03)
*K5. Do you know the preventive measures against COVID-19?*	63.5, 4, <0.01	1.2, 2, 0.55	12.6, 2, <0.01	0.84, 1, 0.36
	(0.43) ***	(0.06)	(0.19) *	(0.05)
*K6. In your opinion, COVID-19 comes from:*	18.3, 4, <0.01	1.1, 2, 0.57	6.8, 2, <0.05	7.3, 1, <0.01
	(0.23) **	(0.06)	(0.140)	(0.145) *
*K7. In your opinion, COVID-19 affects:*	85.9, 8, <0.01	8.5, 4, 0.075	9.4, 4, 0.051	1.5, 2, 0.47
	(0.35) ***	(0.11)	(0.17)	(0.07)
*K8. According to you, compared to malaria, COVID-19 is:*	129.6, 8, <0.01	5.1, 4, 0.24	12.6, 4, <0.05	5.9, 2, 0.052
	(0.43) ***	(0.09)	(0.135) *	(0.13)
*K9.In your opinion, compared to EBOLA, COVID-19 is:*	114.1, 12, <0.01	16.2, 6, <0.05	17.2, 6, <0.01	5.6, 3, 0.14
	(0.33) ***	(0.15) *	(0.16) *	(0.13)
*A1. Can you visit someone who has COVID-19?*	11.9, 4, <0.05	4.6, 2, 0.10	9.1, 2, <0.05	1.1, 1, 0.29
	(0.19) *	(0.16)	(0.16) *	(0.06)
*A2. If it is someone in your family (child, spouse...) who suffers from COVID-19, can you visit him/her?*	11.0, 4, <0.05	11.2, 2, <0.01	11.1, 2, <0.01	0.3, 1, 0.59
	(0.18) *	(0.18) *	(0.18) *	(0.03)
*A3. The people most affected by COVID-19 are adults. Do you think there is a need to take action to prevent COVID-19 infection for children and young adults?*	29.4, 4, <0.01	5.3, 2, 0.070	2.6, 2, 0.27	27.4, 1, <0.01
	(0.29) **	(0.12)	(0.09)	(0.28) **
*A4. Do you agree that COVID-19 will be successfully controlled and defeated in the DRC?*	4.9, 4, 0.294	1.5, 2, 0.47	2.5, 2, 0.29	1.6, 1, 0.21
	(0.119)	(0.07)	(0.09)	(0.07)
*A5. How do you find the origin of COVID-19?*	73.4, 12, <0.01	6.9, 6, 0.33	36,3, 6, <0.01	4.7, 3, 0.20
	(0.27) **	(0.10)	(0.23) **	(0.12)
*A6. During this COVID-19 crisis period, do you eat as before?*	5.4, 4, 0.25	1.2, 2, 0.55	2.3, 2, 0.32	0.7, 1, 0.41
	(0.13)	(0.06)	(0.08)	(0.04)
*A7. What means of transport do you use to get to the market?*	8.7, 4, 0.07	2.4, 2, 0.31	0.5, 2, 0.78	3.017, 1, 0.08
	(0.16)	(0.09)	(0.04)	(0.093)
*A8. How often do you go to the market for supplies or for sale per week?*	3.1, 4, 0.544	0.00, 2, 1.00	1,3, 2, 0.52	3.1, 1, 0.08
	(0.10)	(0.00)	(0.06)	(0.10)
*A9. If someone around you has symptoms of COVID-19, what would you do?*	76.5, 8, <0.01	3.1, 4, 0.535	28.3, 2, <0.01	5.0, 2, 0.08
	(0.33) ***	(0.07)	(0.20) **	(0.12)
*A10. If you think you have COVID-19, whom will you first go to for treatment?*	65.4, 8, <0.01	1.4, 4, 0.844	19.8, 4, <0.01	5.0, 2, 0.08
	(0.31) ***	(0.05)	(0.17) *	(0.12)

Cramer’s V: * weak association; ** average association; *** strong association.

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
