# Peer review of "COVID-19 Pandemic: Knowledge and Attitudes in Public Markets in the Former Katanga Province of the Democratic Republic of Congo"

_ijerph, 2020, doi:10.3390/ijerph17207441_

Round 1
Reviewer 1 Report
This manuscript describes, awareness of the risk of contamination among people frequenting public markets in the former Katanga province in the south of the DRC. The participants was market vendors or customers in three large cities (Kolwezi, Likasi, and Lubumbashi) and one small town (Lwambo) of the former Katanga province. Although the research question and sample collection is quite interesting, I have several major concerns:
Major comments:
Introduction
- The literature review is very general
- In the literature, there are numerous studies on the knowledge and practices about COVID-19 in the population.
Methods:
- The study lacks representativeness. There is no calculation of the sample size, there is no a type of sampling.
- Ethical aspects of the study are not detailed.
- Did they use a standardized questionnaire of the literature?
- Was the questionnaire previously validated and piloted?
Results
- How did you get a sample size of 347 people?
- Table 2. Responses of participants to questionnaire surveys done should be compared among between the 4 groups using Chi square test.
- If the potential participants was market vendors or customers, why were knowledge and attitudes not compared between among market vendors vs. customers?
- Did both groups have the same sociodemographic characteristics and the same knowledge and attitudes?
The presentation of the observational results are incorrect:
- How were the observational analyzes performed?
- How long were sellers and buyers observed?
- What was the distribution of washing devices in the markets?
- According to the authors "nowhere did we observe either sellers or buyers washing their hands after reciprocal contacts". How long was hand washing observed in study participants?
- The results of the use of masks, hand washes, sellers would blow into plastic bags before placing food into the bags, and requesting another plastic bag without prior blowing should be presented in tables or figures.
Discussion
This study has several limitations. It can lead to biases in the findings. The authors do not include or discuss the limitations of the study.
The study is relevant
It is original
It is necessary to review the English style.
Reviewer 2 Report
Comments to the authors:
The manuscript entitlted “COVID-19 Pandemic: Knowledge and attitudes in public markets in the former Katanga province of DR Congo” evaluated the knowledge of COVID-19 disease by people in public markets investigating related behaviors and attitudes. The topic, considering the current epidemiological situation, is extremely interesting, especially for the potential indications that this analysis can provide to the various stakeholders in order to adequately takle the spread of SARS-CoV-2. In this regard, although the Authors have provided some interesting results and potential points for discussion to increase the effectiveness of the actions to combat the pandemic in the African content (and specifically in the Congo), the manuscript has numerous weaknesses which at present prevent its possibility to be published in its current form.
Introduction
The introduction is extremely limited. The authors should make an effort to better and more fully present the rationale of the study, especially with reference to the particular situation of Congo and its national health system. For example, epidemiological data are completely missing and also the presentation of the contrast measures implemented by the Government is frankly very limited and poor. It is not clear to this reviewer why public markets and shops selling food were exempted by lockdown. Moreover, there is no description of the indications about personal protective equipment that people and/or workers should use in public or work places. Also the aim of the study should be better described since in the current version of the manuscript the authors refer only to awareness of the risk of contamination.
Methods
Regarding the methods, and particularly the questionnaire, the Authors should provide more and high quality information in order to have a better understanding of the results obtained. For instance, Has the questionnaire been validated or at least tested in a pilot group in order to check for any problems in understanding the questions by the interviewees? The authors stated that the questionnaire evaluated different dimensions and issues such as clinical characteristics and transmission of SARS-CoV-2, prevention measures of COVID-19 and finally attitudes and behavior related to this disease. In my opinion, it would be important for the Authors to provide a brief and concise description of the questionnaire sections and at least some of its key questions for each section.
Another important question concerns the evaluation methodology with which the interviewers evaluated the implementation of prevention measures. Indeed, the Authors stated that ”…the interviewers observed the markets and the people to obtain information on good practices against the dissemination of SARS-CoV-2 (such as hand washing and masks wearing)”. In this regard, did the interviewers have some sort of check-list? a diary or an event log?
Moreover, what about the personal protective equipment? And the and social distancing?
Also the statistical section should be improved.
Results
The “Results” section should be shortened considering that the findings of the study are also presented in several very large tables. In this section, I suggest to the Authors to focus only on the statistically significant results obtained in the study.
Discussion
It seems to me that the discussion in its current form is essentially a repetition of the results, perhaps slightly more in-depth but offers little food for thought or discussion. Considering the results, I would have expected the authors to use the aforementioned findings to highlight the critical issues that emerged and to take inspiration from these to start devising potential solutions and areas of intervention on which to act to improve people's awareness of SARS-CoV-2. These issues are partially treated in the “Conclusion” section but in a very limited way. I suggest that the Authors strengthen the “Discussion” section by also taking a cue from the numerous papers that have already been published in the literature on the same topic. Indeed, another problem of the discussion is represented by the fact that the results obtained by the authors are not discussed in an international context.
Reviewer 3 Report
The paper describes public perception regarding COVID-19 at an early stage of the pandemic in Africa. The authors' approach is interesting and presents the specificity of the problems and risk factors for the spread of the epidemic in the African country. Certainly, the approach has certain limitations, as the authors pointed out in the "Discussion" section. In my opinion, however, the article deserves to be published after minor corrections, as below:
RESULTS
Table I:
In the case of Kolwezi, Lwambo, and Lubumbashe- percentages for the source of information in total are not 100%. The abbreviation SES was used - this abbreviation should be clarified
“COVID attack team “ and “The support team” – what does it mean in terms of DR Congo in the context of fighting the epidemic? - a short comment is needed, perhaps in the introduction to the work
DISCUSSION
I miss the discussion on the differences - sometimes large- in the responses to questions by people in different cities
Round 2
Reviewer 1 Report
No comments
Author Response
Thank you.
Minor editorial corrections have been made.
Reviewer 2 Report
I acknowledge that the Authors have made a commendable effort to implement the manuscript following the comments and suggestions of this reviewer. in fact, the new version of the article is significantly improved especially in the part of the introduction and the materials and methods. In my opinion there are still some doubts about the discussion of the manuscript which I still find weak and little effective in enhancing the results obtained. The authors stated that "we felt that we should not speculate too much beyond our survey findings, especially in view of the specific circumstances and characteristics of COVID-19 in Sub Saharan Africa" but this reviewer did not ask to carry out free speculations but to draw (from the results) useful indications (to be proposed to all interested stakeholders) to define, plan, implement and carry out preventive and protective tools and strategies to strengthen people's awareness against SARS-CoV2 (taking a cue from already published international studies).
A careful rereading of the text is recommended to correct the many typos (for example see pag.2 line 47, pag.3 lines 109-112...)
Author Response
Thank you for the positive comments.
I our discussion we did comment on the main findings of the survey (generally poor knowledge and awareness, worries about food, sources of information, lack of confidence in the authorities, handwashing devices, "demonic" origin of COVID-19). However, we do not feel that these findings allow us to formulate strong proposals to define, plan, and implement strategies against SARS-CoV-2.
Nevertheless, in our substantially expanded conclusion, we have referred to a recently published document that addresses these required strategies in a remarkable way (Ref #23: National Academies of Sciences, Engineering, and Medicine. 2020. Encouraging Adoption of Protective Behaviors to Mitigate the Spread of COVID-19: Strategies for Behavior Change. Washington, DC: The National Academies Press. https://doi.org/10.17226/25881). Moreover, we have also added a final paragraph of "further considerations" regarding the unanticipated low severity of the COVID-19 in Sub-Saharan Africa (except for South Africa). Admittedly, these "afterthoughts" are not strictly related to the results of the survey, but we feel that they are relevant for those interested in the issue of the pandemic in Africa.